# Redirecting patients from the pediatric emergency department to community locations for care: A qualitative study of healthcare professional and leader perspectives

Erica Qureshi[1,2]*, Quynh Doan[1,2,3], Jessica Moe[4,5,6,7], Steven P. Miller[2,3], Garth Meckler[2,3], Brett Burstein[8,9], Jehannine (J9) Austin[10,11]

1 Faculty of Medicine, University of British Columbia, Vancouver, Canada, 2 British Columbia Children's Hospital Research Institute, Vancouver, Canada, 3 Department of Pediatrics, University of British Columbia, Vancouver, Canada, 4 Department of Emergency Medicine, University of British Columbia, Vancouver, Canada, 5 BC Centre for Disease Control, Vancouver, Canada, 6 Department of Emergency Medicine, Vancouver General Hospital, Vancouver, Canada, 7 Department of Emergency Medicine, BC Children's Hospital, Vancouver, Canada, 8 Montreal Children's Hospital, Division of Pediatric Emergency Medicine, McGill University Health Centre, Montreal, Canada, 9 Department of Biostatistics, Epidemiology and Occupational Health, McGill University, Montreal, Quebec, Canada, 10 Department of Medical Genetics, University of British Columbia, Vancouver, Canada, 11 Department of Psychiatry, University of British Columbia, Vancouver, Canada

* erica.mcdonald@bcchr.ca

## Abstract

### Objectives

Emergency department (ED) to community (ED2C) programs, which redirect patients from the pediatric ED to community healthcare professionals represent a promising strategy to reduce the impact of non-urgent visits on the pediatric ED. Given an ED2C program's potential impact on various care professionals, we completed a qualitative study to explore key informants' attitudes and perceptions of pediatric ED2C programs.

### Methods

We conducted one-on-one semi-structured interviews with key informants in British Columbia, Canada. Participants included: pediatric ED staff – triage nurses and physicians; community professionals – pediatricians and family physicians; and health system leaders responsible for pediatric and emergency care in British Columbia. Interviews were recorded, transcribed verbatim, de-identified, and analyzed using reflexive thematic analysis within an interpretive description framework. A visual model was developed to depict key themes in attitudes and perceptions towards pediatric ED2C programs.

**Data availability statement:** Data (i.e. interview transcripts) cannot be shared publicly to ensure the anonymity of participants, additional illustrative quotes have been included in Supporting Information File 1. Data are available from the University of British Columbia's Research Ethics Board for researchers who meet the criteria for access to confidential data. The UBC Children's & Women's REB can be contacted via email at Manisha.Sangha@bcchr.ca.

**Funding:** The author(s) received no specific funding for this work.

**Competing interests:** The authors have declared that no competing interests exist.

## Results

We interviewed 24 participants: 6 community professionals, 11 pediatric ED professionals, and 7 healthcare leaders. Participants viewed the ED2C program as a valuable solution to address pediatric system strain provided that systemic barriers are addressed, and both emergency and community settings are equipped with adequate training and resources. Participants emphasized the need for clear guidelines on eligibility and operations to build confidence and enhance program effectiveness.

## Conclusions

Our findings suggest there is support for ED2C programs as a means to reduce the impact of non-urgent pediatric ED visits and strengthen community-based care. Successful implementation will require coordinated planning, resource investment, and clear operational frameworks.

## Introduction

Pediatric emergency department (ED) crowding is partly driven by many non-urgent visits, often stemming from fragmented primary care access and caregivers' difficulty navigating and self-triaging a child's health needs [1–4]. Crowding in pediatric EDs adversely affects quality of care, results in longer wait times, a greater proportion of patients who leave without being seen, increased morbidity, and reduced patient satisfaction [5–8]. Addressing the impact of non-urgent visits is crucial; effective solutions have the potential to reduce pediatric ED crowding thereby improving patient experience and quality of care.

Connecting patients with non-urgent healthcare concerns to community health professionals may reduce crowding in the pediatric ED [9]. This has been accomplished through Emergency Department to Community (ED2C) programs, where non-urgent patients are offered a healthcare appointment with a community professional in lieu of waiting for care in the pediatric ED. In a study of families waiting for care in a Canadian urban tertiary pediatric ED, 58% reported an interest in redirection to a community-based family physician or pediatrician [10], suggesting that ED2C programs may be acceptable for patients and their families.

While a pediatric ED2C program is a promising strategy to reduce the impact of non-urgent visits to the pediatric ED [9,11,12], it is essential to gather insights from key informants who would be involved in implementing and/or impacted by these programs [13]. To do so we completed a qualitative assessment of key informants from British Columbia (BC) Canada, including health system leaders responsible for pediatric and emergency care and healthcare professionals. The objective of this study was to explore informants' attitudes and perceptions regarding pediatric ED2C programs.

## Methods

### Procedures

We conducted a qualitative (interpretive description) semi-structured interview-based study to generate a visual model depicting the attitudes and perceptions of key informants. Participants included: pediatric ED staff – pediatric ED triage nurses and physicians, community professionals – pediatricians and family physicians, and health system leaders responsible for pediatric and emergency care. All participants were located in BC, Canada and reflected on the province' single pediatric ED. Ethics approval was obtained from the University of British Columbia's Research Ethics Board (H24-02616).

### Data collection

Using the Consolidated Framework for Implementation Research Interview Guide Tool [14], EQ developed a semi-structured interview guide that included open-ended questions to explore informants' perspectives about pediatric ED2C programs.

We used a combination of purposive, convenience, and snowball sampling to recruit participants. Personalized emails were sent to healthcare leaders by EQ and QD inviting participation. We also hung study posters in staff areas of a Canadian urban tertiary pediatric ED, and distributed posters via healthcare professional email newsletters. Following each interview, participants were asked to recommend colleagues. All participants provided written informed consent prior to their interview. Recruitment began October 3, 2024 and ended March 14, 2025.

Participants completed one-on-one semi-structured interviews via Zoom with EQ. Interviews included open ended questions that explored a participant's attitudes and perceptions regarding pediatric ED2C programs. Data collection and analysis were completed in tandem and the interview-guide was iteratively refined. Interviews were recorded, transcribed verbatim, and checked for accuracy prior to analysis. Qualitative data software (NVivo 14) was used to organize and manage data.

### Analysis

The data analysis team included EQ, JA, and QD, our positionality statements are included in Table 1. Our analytic approach involved reflexive thematic analysis within an interpretive description framework. Interpretive description bridges empirical research with practice and aims to produce findings that are clinically relevant [15,16]. Reflexive thematic analysis provided a flexible yet rigorous analytic structure, allowing for iterative coding and ongoing reflexive engagement [17,18]. This analytic approach facilitated an in-depth understanding of participant perspectives while ensuring the visual model generated had practical relevance.

Analysis was completed by EQ who shared progress with JA following each interview and as results were created.

Table 1. Positionality statements from data analysis team members.

| Research Team Member | Positionality Statement |
|---|---|
| EQ | A White cisgender woman who is an MD/PhD candidate with training in qualitative and quantitative methods and a mother who has taken her child to the pediatric ED |
| JA | A White, agender person with no children who was assigned female sex at birth, who trained clinically as a genetic counselor, and avoids accessing healthcare services – especially emergency departments – due to aversive experiences |
| QD | An Asian cisgender woman, senior pediatric emergency physician, and health services researcher with a leadership role at the BCCH Research Institute |

Interviews were coded line-by-line into basic conceptual units which were then grouped [19]. Groupings were updated iteratively and recursively. EQ and JA met regularly to discuss linkages and inconsistencies among transcripts, memos, and codes, as well as identify illustrative quotes and refine these connections visually. Recognizing the critiques of "saturation" as a recruitment endpoint [20,21], we opted for theoretical sufficiency – prioritizing the information power of our data and adequacy of the constructed model for its envisioned purpose – as our indication to stop data collection [22].

Once EQ and JA believed the visual model was an accurate representation of informants' attitudes and perceptions, we completed member checking with 3 participants of varied professional backgrounds. Based on participant feedback we made small changes to the visual model.

## Results

### Participants

We completed 24 interviews: 6 with community professionals, 11 with pediatric ED professionals, and 7 with healthcare leadership. Some participants had more than one identity (i.e., a healthcare leader overseeing pediatric emergency department operations who formerly worked as a triage nurse), we asked participants to reflect on one role but recognized that their perspective is shaped by their varied experiences. Prior to conducting interviews, EQ worked professionally with three participants and had a personal connection with one. Interviews lasted 27–62 minutes. Additional information about participants is summarized in Table 2. We appraised information power according to Malterud's framework; the specific aim of this study and the high quality of dialogue, which yielded rich, nuanced, and detailed perspectives, contributed to the strong information power of our data [23,24].

### Model

A visual model which depicted participants' attitudes and perceptions about an ED2C program was developed (Fig 1). Each element of the model is discussed in depth below with illustrative quotes to support our results. Additional participant quotes are provided in S1 File.

Table 2. Characteristics of participants, stratified by role.

| | Community Professionals | | Pediatric ED Professionals | | Healthcare Leadership |
|---|---|---|---|---|---|
| | Pediatrician | Family Physician | Physician | Triage Nurse | |
| Number of interviews | 3 | 3 | 5 | 6 | 7 |
| Interview length Average (range) | 47.3 (43-55) | 43.3 (35-55) | 42.4 (29-60) | 39.5 (30-47) | 39.9 (27-62) |
| Age, years Average (range) | 37 (37−37)* | 54 (47-61) | 47.2 (35-67) | 35.7 (30-47) | 52.9 (40-58) |
| Gender, n | | | | | |
| Woman | 1 | 1 | 4 | 6 | 3 |
| Man | 1 | 2 | 1 | 0 | 3 |
| Declined to answer | 1 | 0 | 0 | 0 | 1 |
| Experience, years Average (range) | 7 (7−7)* | 27 (35-55) | 16.4 (6-35) | 8.4 (2-15) | 7.1 (1-15) |

*One pediatrician declined to provide demographic information but noted having 'over 40 years' experience

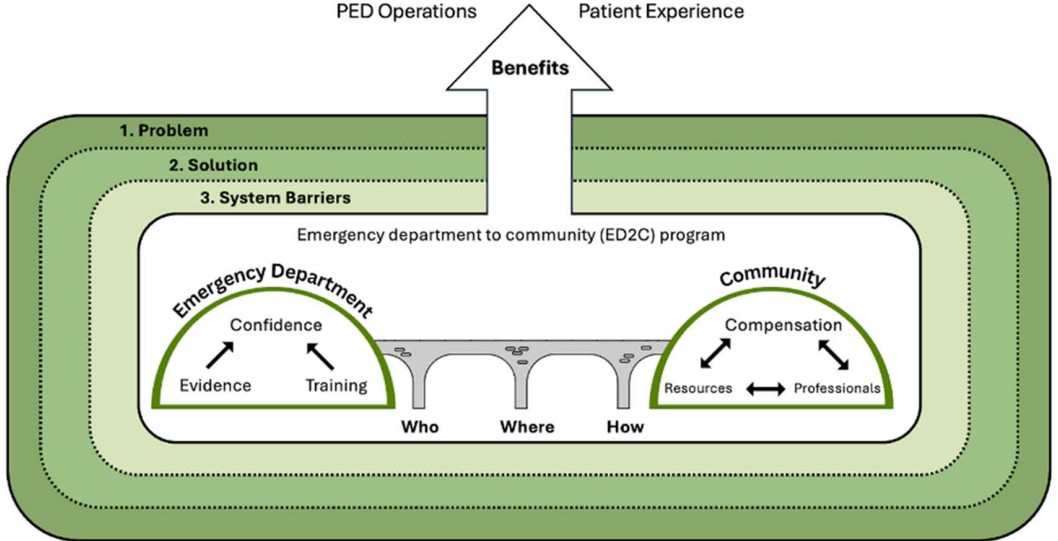

**Fig 1. Visual model of key informants' attitudes and perceptions about a pediatric ED2C program.** Participants considered an ED2C program to be one of multiple potential solutions to systemic challenges in pediatric healthcare. Participants discussed the importance of defining the problem that may be solved by implementing an ED2C program, to determine if there was a need for this solution, and identifying systemic barriers to ED2C implementation. Participants emphasized the importance of ensuring confidence with redirection, with distinct considerations for emergency department and community settings. In the pediatric ED, providing information that the program is evidence-informed, and professional training for redirection was key to building confidence in the program. In the community setting there was a focus on ensuring provision of adequate resources, compensation, and professional capacity to allow for meaningful participation. Participants also underscored some components of a pediatric ED2C program that are essential to create a bridge between pediatric ED and community settings. Participants stressed that for successful implementation, it would be essential to pay careful attention to who would be eligible, where they would receive care, and how the redirection pathway would operate. If these elements were established, participants expressed overall support toward an ED2C program and believed that there would be benefits for both pediatric ED operations and patient experience.

## Problem

Participants acknowledged the growing strain on pediatric ED because of many non-urgent visits and noted this is often driven "because there is a lack of primary care or lack of quick access to care for them so they end up in our EDs." - PED2. Participants believe that many families perceive the emergency department as their entry point into the healthcare system.

Pediatric ED professionals also noted it was difficult to watch non-urgent patients wait when they presented to the pediatric ED because they didn't know about and/or couldn't access other locations for care. One triage nurse stated:

"There's just nowhere to go. It's tough, right when we're watching families stay in our department for like 7 to 10 h, sometimes and they don't really need to be with us." - TN3

While participants thought community connections would be valuable, they raised concerns about redirecting patients to community professionals. In particular some participants worried that shifting care away from the pediatric ED may exacerbate existing strain on the primary care system.

"[We need to make sure that] this [program is] built in a way that isn't perceived as dumping onto a strained capacity. We don't want someone in the community to say you've just taken away a provider from us. We're at a very sensitive time in terms of how we allocate our pediatric workforce across the province." - L6

As a result careful planning is needed to ensure that an ED2C program is valuable rather than burdensome.

## Solution

Participants expressed that many strategies are necessary to reduce the impact of visits for non-urgent concerns and that "anything that we can do to decant from emergency to really operate as a true emergency department and better serve our community is so important." - L7. While an ED2C program was seen as a promising strategy to reduce pediatric ED crowding, participants questioned if preventing visits altogether would be more impactful:

> "This is a model that I think has potential. I think I worry about people having taken the time and made the effort to come into emergency, and whether doing it in that order of emergency to the community will have a slightly lower impact than had they never shown up in the first place. But I think all strategies are needed to address overcrowding." - L5.

Other participants focused on empowering families to seek appropriate care on their own, with one participant stressing: "You want people to not even get to that emergency [department]. Patients need to have some of their own autonomy to take care of themselves." - L3.

These tensions underscored broader questions about program scope, scalability, and quality of care. One informant suggested that an ED2C program should include a longitudinal component as they wondered if "the [health] outcomes would be any different if we're looking at one emergency visit versus one visit through the [redirection] program." - TN6. Others argued that ED2C initiatives should extend beyond pediatric EDs to general EDs.

## System barriers

Participants identified several systemic barriers to implementing an ED2C program, particularly related to funding structures and political priorities. Participants emphasized that while cost savings and improved patient care are compelling incentives, government funding decisions often prioritize short-term political agendas over long-term healthcare efficiencies. As one participant explained:

> "The ministry... are really focused on net budgetary expenses which is unfortunate because it's intuitive and probably the strongest argument is, you know, cost savings. But just being on a political cycle, where our elected officials change and their budgetary priorities change... Unfortunately, the system is not super responsive to that concept of budgetary offsets." - L5.

This disconnect between fiscal policy and healthcare sustainability was echoed by other participants who explained that preventive programs like ED2C struggle to secure funding.

A second critical barrier centered on unintended consequences of redirection. Some participants worried that expedited community access through ED2C might paradoxically increase pediatric ED demand. One clinician cautioned:

> "If we say that a community appointment is this valuable resource that otherwise [patients] are unable to get their hands on then they might say...'I don't know that I need an emergency department, but it was nice to have been seen by a pediatrician last time...and we didn't even have to wait.' " - PED3.

## Emergency department

**Confidence.** Participants emphasized that clarity about program operations and community service capabilities is critical for pediatric ED professional confidence. Knowledge of the community providers involved and that the redirection pathway would result in high quality of care was equally important:

> "For the department knowing who we're sending to and what the capabilities are, because a huge part of the success of something like this is the ability to describe the destination and to say, 'Oh, you know what they're amazing. This is who it is. This is who you're gonna see'... and they can feel confident that experience is gonna be [positive]." - PED1.

In addition to building a care pathway that provides high quality care, breaking the existing silos between acute and community care such that professionals can speak to each other's expertise is essential.

### Evidence

Program safety was a recurring priority "The worst case scenario and the thing that could really... freak everyone out is if you send a kid home and an adverse event happens and they come back and they're very, very unwell." – TN1. To mitigate fears, participants called for transparency about how the program had been developed by "letting nurses know that their feedback has been taken into consideration all throughout the planning and implementation." – TN1. Another key component was communicating if "similar centers have done it successfully, and that it's not just like something that we're trialing out of desperation or like just we made up out of thin air." – TN4.

Finally, guidance from regulatory bodies and explicit institutional support was as essential to alleviate liability concerns and ensure professionals felt supported participating in this program:

> "We always have concerns about putting that onus on [nurses], and I think that it would have to be very clear that you're being directed by the organization to support that service… We can do that with our licensing for sure." – L4.

### Training

Training PED professionals to understand and meaningfully consider the historical and social contexts that shape patient perceptions is important for implementing redirection programs in a thoughtful and effective manner. Without this awareness, redirection may be misinterpreted by some populations as exclusion rather than appropriate care. As one participant emphasized, communication must be:

> "trauma informed because there are also populations who for a number of historical and ongoing reasons have a mistrust of the medical system. And so how might I perceive being sent away from the emergency department differently if I'm part of that population." - P2.

By equipping professionals with the skills to communicate sensitively and explain the intent of redirection, the program is more likely to be perceived as supportive rather than dismissive. For instance, one participant thought that it would be helpful to

> "have some resources available that explain the program because... trust is such a big factor that you know, you say, "oh, go see a doctor". They might think that's like door number two or a tier down, and it's important to reassure [that it isn't]." - L2.

Additionally, feeling supported in redirection could be bolstered by including "another secondary nurse to kind of see that patient and kind of make that final decision of whether they should go or not." - TN3.

Participants noted that there are many junior nurses working in the pediatric ED which may reduce their confidence with redirection. To bolster confidence triage nurses should be reminded they have the beginning of redirection

conversations when patients waiting in the pediatric ED ask if they can leave. Possibly the most important way to foster pediatric ED nurse comfort with redirection is to highlight nurses' existing skills:

> "We determine where you go in the department anyways, that's on the nursing. So we have all the prerequisite skills. It's just understanding what's that other room that we're sending them to like, not the acute, not the fast track, we're sending them to the community." - L4.

## Community

Fair payment for community healthcare professionals was repeatedly cited as essential for incentivizing participation. Participants emphasized that compensation must reflect the work required and account for opportunity costs, given family physicians' competing demands.

Additionally, adequate resources and support systems were deemed critical for ED2C program success. Participants emphasized that comparable community resources (e.g., social workers, expedited specialist referrals) are necessary to match pediatric ED capabilities and ensure family satisfaction. Dedicated support staff to manage community professionals administrative tasks were also deemed valuable such that physicians can focus on patient visits and review tests ordered by other physicians during appointments on previous days.

Comfort with pediatric care emerged as a non-negotiable requirement for community professionals. Participants acknowledged variability in family physicians' pediatric proficiency and stressed the need for targeted recruitment of confident physicians:

> "There's a huge spectrum out there... I wouldn't say, all family doctors are comfortable with managing kids who are sick, potentially. So I think that would be the thing, is to have a group of providers who are willing to see these kids and feel comfortable managing them in the community." - PED2.

While comfort providing care to pediatric patients is essential, participants noted that family physicians comfortable providing pediatric care may be better suited to provide care for patients who are redirected compared to pediatricians:

> "I think, what's interesting is that we see a lot of primary care in the emergency department for which you don't really get training as a pediatrician and so you actually are probably more comfortable dealing with those problems as a family physician than a pediatrician would be." - P2.

## Bridging the ED and community

**Who.** Participants highlighted the importance of clear objective criteria to determine which patient should be eligible for redirection. Inclusion and exclusion criteria should be evidence-informed, developed with a focus on safety, and objective to minimize ambiguity and ensure safety and consistency across professionals. Clear objective eligibility criteria was also viewed favourably by community professionals: "So people can anticipate what type of patients they might be seeing." - P3

**Where.** The location where care is provided plays a significant role in the success of the program. Professionals favoured a child friendly environment to bolster patient satisfaction and comfort. As one participant summarized, "you can't overemphasize the importance of the venue in convincing people that it's an appropriate place for their sick child." - P1

Community professionals also indicated a preference for a clinic location that is distinct from their general practice, especially one with easy access to essential services such as labs and x-ray. Family physicians and pediatricians expressed their comfort providing care in various settings and noted that a location close to testing facilities – for

example co-located at a children's hospital – would streamline care and help patients receive timely and comprehensive evaluations.

### When

Professionals also expressed that the program should be accessible outside of business hours. Offering care outside of the traditional 9-to-5 schedule would increase accessibility for parents and improve the feasibility of the program.

> "The other thing I think you'd have to look at is when it would be open, because we see a lot of patients that are working in entry level jobs, you know, minimum wage. They can't take time off to take their kid to the doctor… to be honest it's not much use to offer yet another 9 to 5 program." - PED4

### Benefits

**Patient experiences.**  The ED2C program has the potential to significantly improve patient experiences. Participants believed the program would help families understand what constitutes an emergency and could reduce future visits to the pediatric ED for non-urgent reasons. Some participants felt this program would reassure patients, particularly those who struggle to determine if care can be delayed or if a visit is a true emergency. Participants also noted that an ED2C program could foster

> "Better relationships between the public and healthcare providers, too. I think when people are sitting in an emergency department for ten hours it is very easy to feel overlooked and like we aren't taking the needs as seriously when it's just the lack of resources and our attention has to be focused on people who are sicker. But I think it could be really hard for people to understand, with no health background." - TN5

This program would also benefit patients who were redirected by removing the stress associated with seeking care. One participant felt an ED2C program

> "Might actually be more therapeutic to see them in an office-based setting, whether that's in hospital or in the community. But somewhere without the stress and anxiety of the emergency room." - L6

### Pediatric ED operations

Participants also believed the program would offer significant advantages in improving efficiency by decreasing the overall patient load in the pediatric ED. This would allow resources to be more focused on emergent cases, enhancing professionals capacity to address urgent needs in a timely fashion. Pediatric ED physicians noted that these benefits would be valuable as:

> "[Long wait times are] a bit soul destroying, because, you know, you start at 9 o'clock, and the 1st thing you do is you go in and see the 8 h waits...and then it takes longer because they've waited 8 hours so they want tests, they want blood work. And so the kid that you might have been able to see in 2 hours and just get out all of a sudden becomes quite a lot of discussion and education." - PED4.

Triage nurses also highlighted that redirection may reduce distractions as some families waiting in the pediatric ED "are constantly at triage interrupting and, wanting things asking, 'can I go can I go?'" - TN3

Participants felt this program could strengthen the role of community providers as trusted sources of care by extending the strong reputation of children's hospitals into community settings. By decentralizing pediatric care, the program encourages greater collaboration and trust across the healthcare system – enhancing care both within and beyond specialized pediatric hospitals.

Overall, participants felt an ED2C program would address both crowding and care fragmentation. As one participant summarized:

> "Considering the healthcare or human health care resource crisis and limitations that we have, anything we can do to restructure the patients that we see in the emergency department and redirect others out into the community would only help us see those patients that are sick that require urgent or emergent services." - L1.

## Discussion

This work addresses a critical gap in the literature by capturing the voices of those who would operationalize and benefit from an ED2C program. Through this study we explored the perceived need for, interest in, anticipated barriers and facilitators to implementing, and potential benefits of a pediatric ED2C program in a Canadian PED. In general, healthcare professionals and leaders viewed the ED2C program as a valuable response to increasing non-urgent visits. While participants expressed general support for the program, they raised important concerns around primary care capacity, appropriate eligibility criteria, professional confidence, and implementation logistics. Carefully considering our results when designing future ED2C programs will increase the likelihood that ED2C programs are adopted, implemented, effective, and sustained.

Participants pointed to a breakdown in both the availability and public confidence in longitudinal, timely primary care as key factors driving non-urgent use of the pediatric ED. This aligns with data showing that not all Canadians have a primary care professional and for those who do there are barriers to timely access [25,26]. Participants described how, in the absence of accessible primary care, pediatric EDs have become the default entry point to the pediatric health system—a trend that ED2C programs aim to counterbalance.

A striking insight raised by participants was the potential for ED2C programs to catalyze broader shifts in how families engage with the pediatric healthcare system. While redirection offers clear benefits to both patient experience and pediatric ED operations, it may also help re-establish accessibility to community professionals as trusted and competent sources of care. This, in turn, could address a foundational driver of pediatric ED overcrowding: families' preference for pediatric EDs based on the perceived superiority of the care in that setting [1,10,27]. If families experience positive, competent care following redirection, it may create a ripple effect, gradually reshaping their future care-seeking behaviours. In this way, ED2C programs could function not only as an acute response to crowding but also a long-term solution to reduce non-urgent pediatric ED visits.

While participants were optimistic about the ED2C program overall, they emphasized that success would hinge on adequate resources and coordination across settings. This is consistent with literature that concludes coordination across the healthcare system is needed to reduce pediatric ED crowding [7,12,28]. Participants also stressed that primary care redirection should not be offloaded onto already overstretched family physicians without structural investment. This echoes the concerns raised in Ontario's evaluation of EDs which cited insufficient community capacity and a lack of coordination across the healthcare system as barriers to redirection [29]. Ensuring that community and pediatric ED perspectives – as we have done in this study – will be essential at all stages of planning and implementation of an ED2C program.

Participants also emphasized the need for clear, objective eligibility criteria to ensure comfort offering redirection, patient safety, and equity. Previous studies redirecting pediatric patients reported children returning to the

pediatric ED and requiring admission to hospital [30] – a fear raised by participants within this study. This highlights the need for future research to develop evidence-informed eligibility criteria in collaboration with community professionals; work our team is currently completing. Participants also noted that clear eligibility criteria would help ensure decisions to offer redirection are not subjective. Since patients with visible minority features are systematically assigned lower acuity scores and face longer wait times due to language and advocacy barriers [31–33], ensuring that criteria for redirection are not subject to these disparities is essential. Training in trauma-informed communication was also described as essential to foster professional comfort and promote patient trust and buy-in to redirection.

Further, families facing structural barriers—e.g., concerns with transportation, work schedules, or childcare—may rely on the pediatric ED not only for medical care but also for its reliability and predictability [3,27,34]. Allowing families to choose redirection – rather than dictating that those eligible must be redirected – acknowledges that a community healthcare appointment may not be preferred for some families and is another way that equity can be preserved within an ED2C program.

While most family physicians receive some pediatric training, studies suggest many lack confidence in managing acute pediatric presentations, particularly in infants or children with complex needs [35–37]. This was echoed as participants stressed that some family physicians may not be comfortable managing pediatric patients. However, the consistent message from community participants was pediatricians may not be the most appropriate to manage primary-care level pediatric concerns as they are trained to provide care for complex concerns. Overall, participants agree that family physicians who are comfortable and have experience supporting pediatric patients would be the best community professionals to participate in a pediatric ED2C program.

This study adds to the limited literature exploring professional perspectives on pediatric ED redirection programs. By focusing on the concerns of those most likely to deliver or interface with the program, our results offer concrete guidance for program design and implementation. Participants were asked to think about an ED2C program within one pediatric ED; as a result, some of our results may be specific to the particular local context within Canada. Further, participants were purposively sampled, which may have introduced selection bias toward those more engaged or opinionated about an ED2C program. As a qualitative study, findings are based on perceptions and may not fully predict behavior during implementation. Nevertheless, the diversity of participants' roles enhances the trustworthiness and robustness of insights.

## Conclusion

This qualitative study suggests that an ED2C program could play a valuable role in mitigating non-urgent pediatric ED visits and strengthening community care pathways. However, success depends on thoughtful implementation that considers professional capacity, equity, communication, and infrastructure. Designing an ED2C program that is practical, trusted, and effective will require collaboration across the continuum of care.

## Supporting information

**S1 File. Additional Illustrative Quotes.**
(DOCX)

**S2 File. COREQ Checklist.**
(PDF)

## Acknowledgments

We thank all participants for their thoughtful contributions to this work.

## Author contributions

**Conceptualization:** Erica Qureshi, Quynh Doan, Jessica Moe, Steven P Miller, Garth Meckler, Brett Burstein, Jehannine (J9) Austin.

**Data curation:** Erica Qureshi, Jehannine (J9) Austin.

**Formal analysis:** Erica Qureshi, Jehannine (J9) Austin.

**Investigation:** Erica Qureshi, Quynh Doan, Jehannine (J9) Austin.

**Methodology:** Erica Qureshi, Quynh Doan, Jessica Moe, Steven P Miller, Garth Meckler, Brett Burstein, Jehannine (J9) Austin.

**Project administration:** Erica Qureshi, Quynh Doan, Jehannine (J9) Austin.

**Supervision:** Quynh Doan, Jehannine (J9) Austin.

**Visualization:** Jehannine (J9) Austin.

**Writing – original draft:** Erica Qureshi, Quynh Doan, Jehannine (J9) Austin.

**Writing – review & editing:** Erica Qureshi, Quynh Doan, Jessica Moe, Steven P Miller, Garth Meckler, Brett Burstein, Jehannine (J9) Austin.

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
