## [Decision Letter · Decision Letter 0]

20 Oct 2025

Dear Dr. Qureshi,

Thank you for submitting your manuscript to PLOS ONE. After careful consideration, we feel that it has merit but does not fully meet PLOS ONE’s publication criteria as it currently stands. Therefore, we invite you to submit a revised version of the manuscript that addresses the points raised during the review process.

We look forward to receiving your revised manuscript.

Kind regards,

Moises Auron, MD, FAAP, FACP, SFHM, FRCP (Lon), FRCPCH

Academic Editor

PLOS ONE

Journal Requirements:

Reviewers' comments:

Reviewer's Responses to Questions

**Comments to the Author**

1. Is the manuscript technically sound, and do the data support the conclusions?

Reviewer #1: Yes

Reviewer #2: Yes

2. Has the statistical analysis been performed appropriately and rigorously?

Reviewer #1: Yes

Reviewer #2: Yes

3. Have the authors made all data underlying the findings in their manuscript fully available?

Reviewer #1: No

Reviewer #2: Yes

4. Is the manuscript presented in an intelligible fashion and written in standard English?

Reviewer #1: Yes

Reviewer #2: Yes

Reviewer #1: This article provides great qualitative insight from relevant caregivers/stakeholders about important factors to consider when implementing a program to transition qualifying pediatric patients from the emergency room to a community provider to better address both local staffing/resources and patient experience/quality of care. It presents an important first step to help build a successful program.

Not all of the interview data is made available, save for the specific excerpts that are quoted directly in the results portion. I do think this makes sense in the context of this article. However, calling this out would be helpful for the reader to understand why only portions of the responses are being shared or why certain responses are being withheld (was there personal information about the interviewee, was it repetitive or consistent with other opinions?). It would also add value to your analytic section to further explain how you parsed through the interviews for relevant data. I would also understand not providing all the transcripts given these interviews were lengthy and would be a significant amount of data to present in entirety. Perhaps only sharing the 'iterative groupings' rather than complete transcripts as an additional resource would be helpful.

The study provided the demographic information (race, gender, education, emergency room connection) for 3 individuals involved in the study (within the methods section). As it is presented, it is not clear why this information is relevant.

Reviewer #2: Overall, this was a good study, aiming to demonstrate that there are ways to successfully reduce busy, emergency department volumes for low acuity patients. Although the results of this study were not surprising, it was a good summarization. There may be some bias given the small amount of people that were interviewed The study was very well written and easy to read and to interpret.

**Do you want your identity to be public for this peer review?** For information about this choice, including consent withdrawal, please see our Privacy Policy

Reviewer #1: **Yes: ** Noah Schwartz, MD

Reviewer #2: **Yes: ** Lauren N. Mientkiewicz

---

## [Author Response · Author response to Decision Letter 1]

24 Nov 2025

Journal Requirements:

We have reviewed the style requirements and made edits accordingly to ensure our submission is aligned with the requirements.

There are ethical restrictions with sharing the data as participants did not consent to have the recorded and transcribed conversations shared. Aligned with Reviewer 1’s feedback below we have shared iterative groupings of additional illustrative quotes in a supporting information file for readers. Additionally, complete data can be requested from the research ethics board that approved this study (our data availability statement in the submission form has been revised to note this).

This has been completed.

This has been completed, one small grammatical change was made to one citation.

Reviewers' comments:

Reviewer's Responses to Questions

Comments to the Author

1. Is the manuscript technically sound, and do the data support the conclusions?

Reviewer #1: Yes

Reviewer #2: Yes

2. Has the statistical analysis been performed appropriately and rigorously?

Reviewer #1: Yes

Reviewer #2: Yes

3. Have the authors made all data underlying the findings in their manuscript fully available?

Reviewer #1: No

As noted above we have included additional illustrative quotes in a supporting information file and updated our data availability statement.

Reviewer #2: Yes

4. Is the manuscript presented in an intelligible fashion and written in standard English?

Reviewer #1: Yes

Reviewer #2: Yes

5. Review Comments to the Author

Reviewer #1: This article provides great qualitative insight from relevant caregivers/stakeholders about important factors to consider when implementing a program to transition qualifying pediatric patients from the emergency room to a community provider to better address both local staffing/resources and patient experience/quality of care. It presents an important first step to help build a successful program.

Not all of the interview data is made available, save for the specific excerpts that are quoted directly in the results portion. I do think this makes sense in the context of this article. However, calling this out would be helpful for the reader to understand why only portions of the responses are being shared or why certain responses are being withheld (was there personal information about the interviewee, was it repetitive or consistent with other opinions?).

We have provided additional information to explain how illustrative quotes were selected in the data analysis section of our methods. Additional quotes would be both lengthy and repetitive as perspectives were consistent. However, for those interested we have included additional illustrative quotes in a supporting information file.

It would also add value to your analytic section to further explain how you parsed through the interviews for relevant data. I would also understand not providing all the transcripts given these intermviews were lengthy and would be a significant amount of data to present in entirety. Perhaps only sharing the 'iterative groupings' rather than complete transcripts as an additional resource would be helpful.

Thank you for this suggestion, as noted above we do not have participants consent to share the full transcripts and they are quite lengthy. However, we did take this suggestion and have provided iterative groupings of additional illustrative quotes in a supporting information file.

The study provided the demographic information (race, gender, education, emergency room connection) for 3 individuals involved in the study (within the methods section). As it is presented, it is not clear why this information is relevant.

We have clarified this in our manuscript by moving the positionality statements into a table and explaining that they are provided for the three research team members who supported our data analysis. Given that reflexive thematic analysis involves the researchers as a ‘tool’ in the research process our unique worldview (as shaped by our gender, race, education, emergency room connection) is important to reflect on and disclose.

Reviewer #2: Overall, this was a good study, aiming to demonstrate that there are ways to successfully reduce busy, emergency department volumes for low acuity patients. Although the results of this study were not surprising, it was a good summarization. There may be some bias given the small amount of people that were interviewed The study was very well written and easy to read and to interpret.

Thank you for this comment. We stopped data collection when we reached theoretical sufficiency (that is when we had enough information to answer our research question in a clinically meaningful way). The goal of this work was not to hear all possible perspectives but instead to gather ‘enough’ to create a visual model that summarized the components of a pediatric redirection program that should be considered before implementation.

---

## [Editor Report · Decision Letter 1]

26 Nov 2025

Redirecting patients from the pediatric emergency department to community locations for care: A qualitative study of healthcare professional and leader perspectives

PONE-D-25-36467R1

Dear Dr. Qureshi,

We’re pleased to inform you that your manuscript has been judged scientifically suitable for publication and will be formally accepted for publication once it meets all outstanding technical requirements.

Kind regards,

Moises Auron, MD, FAAP, FACP, SFHM, FRCP (Lon), FRCPCH

Academic Editor

PLOS ONE
---

## [Editor Report · Acceptance letter]

PONE-D-25-36467R1

PLOS ONE

Dear Dr. Qureshi,

I'm pleased to inform you that your manuscript has been deemed suitable for publication in PLOS ONE. Congratulations! Your manuscript is now being handed over to our production team.

Kind regards,

on behalf of

Dr. Moises Auron

Academic Editor

PLOS ONE